# Oral Administration of *Lactobacillus sakei* CVL-001 Improves Recovery from Dextran Sulfate Sodium-Induced Colitis in Mice by Microbiota Modulation

**DOI:** 10.3390/microorganisms11051359

**Published:** 2023-05-22

**Authors:** Dong-Yeon Kim, Tae-Sung Lee, Do-Hyeon Jung, Eun-Jung Song, Ah-Ra Jang, Ji-Yeon Park, Jae-Hun Ahn, In-Su Seo, Seung-Ju Song, Yeong-Jun Kim, Yun-Ji Lee, Yeon-Ji Lee, Jong-Hwan Park

**Affiliations:** 1Laboratory Animal Medicine, Animal Medical Institute, College of Veterinary Medicine, Chonnam National University, 77 Yongbong-ro, Buk-gu, Gwangju 61186, Republic of Korea; pico9317@naver.com (D.-Y.K.);; 2Nodcure, Inc., 77 Yongbong-ro, Buk-gu, Gwangju 61186, Republic of Korea

**Keywords:** inflammatory bowel disease, *Lactobacillus sakei* CVL-001, anti-inflammation, epithelial barrier, microbiota

## Abstract

Inflammatory bowel disease (IBD) is an intestinal chronic inflammatory disease, and its incidence is steadily increasing. IBD is closely related to the intestinal microbiota, and probiotics are known to be a potential therapeutic agent for IBD. In our study, we evaluated the protective effect of *Lactobacillus sakei* CVL-001, isolated from *Baechu* kimchi, on dextran sulfated sodium (DSS)-induced colitis in mice. The oral administration of *L. sakei* CVL-001 according to the experimental schedule alleviated weight loss and disease activity in the mice with colitis. Furthermore, the length and histopathology of the colon improved. The expression of tumor necrosis factor (*TNF*)*-α* and interleukin *(IL)-1β* genes decreased in the colons of mice that were administered *L. sakei* CVL-001, whereas that of *IL-10* increased. The expressions of genes coding for E-cadherin, claudin3, occludin, and mucin were also restored. In co-housed conditions, *L. sakei* CVL-001 administration did not improve disease activity, colon length, and histopathology. Microbiota analysis revealed that *L. sakei* CVL-001 administration increased the abundance of microbiota and altered *Firmicutes*/*Bacteroidetes* ratio, and decreased *Proteobacteria*. In conclusion, *L. sakei* CVL-001 administration protects mice from DSS-induced colitis by regulating immune response and intestinal integrity via gut microbiota modulation.

## 1. Introduction

Inflammatory bowel disease (IBD) is a chronic, relapsing condition of the gastrointestinal tract that is rapidly increasing in prevalence in many countries [1]. Although the precise causes of IBD have not yet been identified, complex factors such as gut microbiota, genetic predisposition, excessive immune response, and environmental factors are considered to be associated with IBD development [2]. Conventional treatment strategies, including anti-inflammatory drugs, antibiotics, and colectomy, can lead to remission of symptoms in patients with IBD. However, they are not suitable for long-term therapy, and various side effects have been reported [3,4]

The development of next-generation sequencing technology has recently led to extensive research on microbiota. Gut microbiota play a crucial role in protecting the host by influencing nutrition, metabolism, and immune response by maintaining homeostasis [5]. Dysbiosis, which is defined as an imbalance of the composition and function of gut microbiota, has been linked to several diseases, including gastrointestinal, metabolic, and neurological disorders [6]. A significant relationship has also been reported between IBD and gut microbiota [7]. For example, patients with IBD demonstrate significantly reduced microbial diversity [8]. The microbiota’s composition in patients with IBD is characterized by an altered abundance of *Bacteroidetes* and *Firmicutes* and an increased abundance of *Proteobacteria,* compared with those in healthy individuals [9]. Additionally, patients with ileal Crohn’s disease demonstrate a microbial composition with an increased population of *Escherichia* spp. strains, which is characterized by epithelial invasion of the intestinal mucosal surface [10]. These findings imply that the gut microbiota of patients with IBD can influence disease onset and progression. Therefore, modulation of gut microbiota of patients with IBD is an emerging novel therapeutic strategy.

Probiotics are micro-organisms that have salutary effects on the host’s gut microbiota [11]. *Bifidobacterium*, *Lactobacillus*, and *Enterococcus* are representative probiotics that protect against various immune diseases, including IBD, multiple sclerosis, and allergy [12,13]. For example, *Lactobacillus* spp. are beneficial in treating gastrointestinal diseases, such as ulcerative colitis, by competing with intestinal pathogens, maintaining intestinal epithelial barrier function, and immunomodulation [14]. However, the results of in vitro and animal disease experiments differ depending on the characteristics of the *Lactobacillus* strain [15]. Therefore, it is valuable to identify strains with good efficacy and to validate them by applying them to animal disease models.

In the present study, we investigated the protective effects of *Lactobacillus sakei* CVL-001 isolated from *baechu* kimchi, which is a traditional Korean food, against IBD using a mouse model of dextran sulfate sodium (DSS)-induced colitis.

## 2. Materials and Methods

### 2.1. Isolation and Identification of L. sakei CVL-001

*L. sakei* CVL-001 (GenBank OP936132) was isolated from *baechu* (Chinese cabbage) kimchi. The kimchi lysate was inoculated onto deMan, Rogosa, and Sharpe agar (BD Biosciences, San Jose, CA, USA) plates containing 2% calcium carbonate (*w*/*v*). The *L. sakei* CVL-001 isolate was characterized based on Bergey’s Manual of Systematic Bacteriology. The 16S rRNA gene was amplified through a polymerase chain reaction (PCR) using the universal primers 27F (5′-AGAGTTTGATCCTGGCTCAG-3′) and 1492R (5′-GGATACCTTGTTACGACTT-3′). Species identification was confirmed using a nucleotide sequence homology test, which was performed using the Blast program (http://www.ncbi.nlm.nhi.gov (accessed on 11 January 2019)) for registered information (GenBank database). Bacterial cultures for the experiments were obtained by incubating a single colony overnight in 10 mL MRS broth (BD Biosciences, San Jose, CA, USA) at 30 °C with shaking at 150 rpm. Subsequently, a 1:10 dilution of the bacterial suspension was obtained by shaking until the optical density at 600 nm was 0.6, corresponding to 0.89 × 10^9^ colony-forming units (CFU)/mL.

### 2.2. Animals

Six-week-old female C57BL/6 mice were purchased from Damul Science (Daejeon, Republic of Korea). The mice were housed under 12 h/12 h light/dark cycles at a temperature and humidity of 24 ± 2 °C and 50 ± 5%, respectively. Sterilized feed and water were provided ad libitum. All animal studies were reviewed and approved by the Institutional Animal Care and Use Committee of Chonnam National University (approval number: CNU IACUC-YB-2019-72, CNU IACUC-YB-2022-68).

### 2.3. In Vivo Experiments

We performed two independent animal experiments. After an acclimatization period of 1 week, the 7-week-old mice were randomly assigned to six groups in an *L. sakei* CVL-001 efficacy experiment, and five groups in a co-housing experiment (*n* = 9/group). The oral administration was initiated 10 days before DSS treatment; i.e., 10^7^, 10^8^, and 10^9^ CFU *L. sakei* CVL-001 in 200 μL phosphate-buffered saline (PBS) was orally administered daily until the end of the experiment in the *L. sakei* CVL-001 administration groups. In the groups that did not receive *L. sakei* CVL-001, 200 μL PBS was orally administered daily. Colitis was induced in the mice by providing sterile tap water containing 2% DSS (*w*/*v*) (MP Biomedicals^TM^, Santa Ana, CA, USA) from day 0 to day 6 in the DSS-treatment group; the non-DSS groups received plain tap water from day 0 to day 6. In the co-housing experiment, two distinct groups (administered 10^9^ CFU of *L. sakei* CVL-001 and PBS) were housed in a single cage and were referred to as the co-housed groups. As control groups, the groups receiving *L. sakei* CVL-001 and PBS were housed separately in each cage and referred to as the separated groups.

After sacrificing the mice, four were used for histological assessment of colon lesions, and the remaining for immunologic and microbiota analyses. Body weight and clinical sign scores were evaluated daily after DSS treatment. Clinical signs were evaluated by scoring weight loss, stool consistency, and presence of blood in the feces [16]. The detailed scoring system for clinical signs is presented in Table 1.

### 2.4. Histological Evaluation

The colon tissues were fixed for 48 h in 10% neutral buffered formalin and embedded in paraffin. Subsequently, 4 μm sections were prepared for hematoxylin and eosin (H&E) and Alcian blue–periodic acid Schiff (AB-PAS) staining. Histology was evaluated by scoring for the damage to epithelial integrity and inflammatory cell infiltration [16]. The detailed scoring system for histologic evaluation is described in Table 2. AB-PAS-positive goblet cells were counted in five random areas of a section from each mouse.

### 2.5. Real-Time PCR (qPCR)

Easy Blue^®^ kits (iNtRON Biotechnology, Seoungnam, Republic of Korea) were used to extract RNA from the colon. A total of 1000 ng RNA was transcribed into cDNA using the ReverTra Ace™ qPCR RT kit (TOYOBO, Osaka, Japan). Real-time PCR was performed using cDNA as a template with QGreen 2X qPCR Master Mix (Cellsafe, Suwon, Republic of Korea). The primer sequences are described in Table 3. Real-time PCR data were normalized using glyceraldehyde 3-phosphate dehydrogenase (GAPDH) as the housekeeping gene. In this study, real-time PCR was performed using a CFX Connect™ Real-Time PCR System (Bio-Rad, Hercules, CA, USA) using a two-step procedure (40 cycles at 95 °C for 10 s and then at 60 °C for 30 s).

### 2.6. Sequencing of Bacterial DNA Extracted from Mouse Feces

Bacterial DNA was extracted from mouse feces using the PowerMax soil DNA isolation kit (MO BIO, Carlsbad, CA, USA). Illumina 16S Metagenomic Sequencing Library protocols were used to amplify the V3 and V4 regions of bacterial 16S rRNA. DNA quality was evaluated using PicoGreen and NanoDrop. PCR amplification of 10 ng genomic DNA was conducted using the 341F (5′-CCTACGGGNGGCWGCAG-3′) and 806R (5′-GACTACHVGGGTATCTAATCC-3′) primers. The final product was quantified by utilizing KAPA Library Quantification kits (KAPA Biosystems, Wilmington, MA, USA) for Illumina Sequencing platforms and qualified using a LabChip GX HT DNA high sensitivity kit (PerkinElmer, Waltham, MA, USA). Paired-end sequencing was performed on the MiSeq™ platform (Illumina, San Diego, CA, USA).

### 2.7. Operational Taxonomic Unit (OTU) Analysis for MiSeq

OTU analysis was conducted by Macrogen Inc. (Seoul, Republic of Korea). Each MiSeq raw data point was classified by sample to create a FASTQ file, which was processed with adapter sequence removal and error correction using the fastp program. Paired-end data for each sample were assembled into one sequence using FLASH, version 1.2.11. Sequences were clustered based on similarity to form a species-level OTU. For the representative sequence of each OTU, taxonomic assignment was performed with the organism having the highest similarity using BLAST + (v2.9.0) in the Reference DB (NCBI 16S Microbial). A comparative analysis of the various microbial communities was performed using QIIME, version 1.9. To confirm the species diversity and uniformity of the microbial community in the sample, the Shannon index and inverse Simpson index were obtained, and alpha diversity information was confirmed based on the rarefaction curve and Chao1 value. Additionally, beta diversity among the samples was calculated based on the UniFrac distance.

### 2.8. Statistical Analysis

The statistical significance of differences between the groups was determined by a two-way analysis of variance (ANOVA) and a Mann–Whitney U test. PERMANOVA was used to determine significant differences upon beta-diversity analysis. *p* values < 0.05 were considered statistically significant.

## 3. Results

### 3.1. Oral Administration of L. sakei CVL-001 Ameliorates DSS-Induced Colitis

To determine the potential protective effects of *L. sakei* CVL-001 against DSS-induced colitis, mice were administered various *L. sakei* CVL-001 doses according to the experimental schedule shown in Figure 1A. DSS-treated groups exhibited body weight loss and clinical signs, such as diarrhea and bloody feces (Figure 1B,C). Administration of *L. sakei* CVL-001 improved the loss of body weight and disease activity in a dose-dependent manner, and there was a significant difference between phenotypes of the PBS-treated group and those of the group treated with a high dose (10^9^ CFU) of the bacterium among the DSS-treated mice (Figure 1B,C). Moreover, *L. sakei* CVL-001 administration helped recover the shortened colon length in mice (Figure 1D,E), which is a characteristic feature of DSS-induced colitis [17].

Histopathological analysis revealed that compared with the non-DSS groups, those that received DSS treatment exhibited increased infiltration of inflammatory cells into the lamina propria and disruption of the colon’s epithelial layer (Figure 2A,B). Furthermore, the histopathologic severity induced by DSS treatment was ameliorated by *L. sakei* CVL-001 administration in a dose-dependent manner. Additionally, the PBS-treated group and the group treated with 10^9^ CFU of *L. sakei* CVL-001 among the DSS-treated mice exhibited a significant difference (Figure 2A,B). AB-PAS staining was performed to investigate epithelial barrier function based on mucus secretion ability. Compared with the PBS-treated group, the groups that were administered 10^7^ and 10^9^ CFU of *L. sakei* CVL-001 exhibited a significantly higher number of mucus-secreting cells (Figure 2C,D). These results suggest that the oral administration of *L. sakei* CVL-001 alleviates DSS-induced colitis in mice.

### 3.2. Anti-Inflammatory Effect of L. sakei CVL-001 against DSS-Induced Colitis

To evaluate the effect of *L. sakei* CVL-001 on inflammatory responses in the colon, mRNA expression levels were measured. The mRNA expression levels of cytokines such as tumor necrosis factor *(TNF)-α*, interleukin *(IL)-6*, *IL-1β*, and *IL-10* were upregulated in the DSS-treated groups compared with those in the non-DSS groups (Figure 3A–D). However, the mRNA expression of pro-inflammatory cytokines, such as *TNF-α* and *IL-1β,* were significantly downregulated by *L. sakei* CVL-001 administration in the 10^7^ and 10^9^ CFU groups (Figure 3A,C). Mean levels of *IL-6* expression were also lower in groups that received 10^7^ and 10^9^ CFU of *L. sakei* CVL-001; however, the difference was not significant (Figure 3B). On the contrary, the mRNA expression levels of *IL-10*, which is typically known as an anti-inflammatory cytokine, were more significantly upregulated in the 10^8^ and 10^9^ CFU groups than they were in the PBS-treated group (Figure 3D). These results indicate that the administration of *L. sakei* CVL-001 mitigates intestinal inflammation by regulating pro- and anti-inflammatory cytokine expression.

### 3.3. L. sakei CVL-001 Improves Intestinal Mucosal Barrier Function

Various probiotics can prevent intestinal diseases by improving intestinal barrier function [18,19]. Accordingly, we further examined the effect of *L. sakei* CVL-001 on intestinal barrier function by measuring gene expression levels of adherens junction proteins (E-cadherin), epithelial tight junction proteins (claudin-3, occludin, and zonula occludens [ZO]-1), and mucin (MUC) 2. The mRNA expression levels of the genes coding for E-cadherin, claudin-3, occludin, and ZO-1 proteins were lesser in the DSS-treated groups than in the non-DSS groups (Figure 4A–D). However, the levels of genes coding for E-cadherin, claudin-3, and occludin were significantly restored by administration of 10^9^ CFU of *L. sakei* (Figure 4A–C). The mean level of *ZO-1* gene expression was also increased via administration of 10^9^ CFU *L. sakei* CVL-001, although the difference was not significant (Figure 4D). To evaluate mucus-secreting ability, we measured the mRNA expression levels of *MUC2*, which were reduced in DSS-treated mice, but restored by administration of 10^9^ CFU of *L. sakei* CVL-001 (Figure 4E). These findings suggest that the administration of *L. sakei* CVL-001 may alleviate ulcerative colitis by improving intestinal barrier function.

### 3.4. Effect of Co-Housing on DSS-Induced Colitis in Mice Receiving L. sakei CVL-001

Probiotics protect against various diseases, including IBD, by modulating gut microbiota [20]. To elucidate the protective effect of *L. sakei* CVL-001 against IBD through microbiota modulation, a microbiota transfer study was conducted via performing a co-housing experiment. In the separated condition, *L. sakei* CVL-001 administration aided significantly in the recovery of body weight loss and reduced disease activity during the later stages of the experiment (Figure 5A,B). Moreover, reduced colon length and colonic pathology caused by DSS-induced colitis was ameliorated by *L. sakei* CVL-001 administration (Figure 5C–F). However, the PBS and *L. sakei* CVL-001 treatment groups did not exhibit a significant difference in body weight loss (Figure 5A), disease activity index (Figure 5B), colon length (Figure 5C,D), and histopathology score under co-housing conditions (Figure 5E,F). mRNA expression levels of inflammatory cytokines (IL-1β and IL-10) genes coding for intracellular junction (E-cadherin, Claudin3, and ZO-1) that showed significant differences between separated groups did not exhibit significant differences under co-housing conditions (Appendix A). These results suggest that the gut microbiota alteration induced by *L. sakei* CVL-001 administration might have influenced the DSS-induced colitis phenotypes.

### 3.5. Administration of L. sakei CVL-001 Leads to Compositional Change of Gut Microbiota

To identify microbiota alterations, the abundance and composition of gut microbiota were analyzed through OTU analysis using the 16S bacterial rRNA gene. Alpha diversity analysis revealed that *L. sakei* CVL-001 administration led to increased mean levels of alpha-diversity metrics on PD_whole_tree, Chao1, and observed_OTUs, which related to the abundance of gut microbiota in DSS-treated mice, regardless of co-housing; however, there was no statistically significant difference (Figure 6A). The alpha diversity of PBS-treated mice in the co-housed group was also at slightly higher mean levels than that of the separated group (Figure 6A). Principal coordinate analysis (PCoA) was performed to quantify the compositional difference of the microbiota among the groups. In the separated groups, clusters in the PBS-treated group were distinct from those in the *L. sakei* CVL-001-treated group (*p* = 0.01), suggesting that *L. sakei* CVL-001 administration can significantly alter the structure of gut microbiota in unweighted UniFrac distance (Figure 6B). In contrast, the two co-housed groups showed similar clusters (*p* = 0.26), and were relatively similar to those of the *L. sakei* CVL-001-treated groups in separate cages (Figure 6B).

We analyzed the relative abundance of gut microbiota at the phylum and genus levels. The results showed that the abundance of *Bacteroidetes* increased because of *L. sakei* CVL-001 administration, whereas that of *Firmicutes* reduced, regardless of co-housing (Figure 6C). Moreover, *L. sakei* CVL-001 administration reduced the abundance of *Proteobacteria,* regardless of co-housing. We measured the relative abundance of *Escherichia* spp. and *Proteus* spp. at the genus level, and found that these were reduced because of *L. sakei* CVL-001 administration, regardless of co-housing (Figure 6C).

## 4. Discussion

Lactic acid bacteria (LAB) are known to protect against gastrointestinal diseases through various mechanisms. They modulate the host’s immune response by regulating the production of cytokines by interacting with pattern-recognition receptors, such as Toll-like receptor (TLR) 2, expressed on host cells [21,22]. LAB can also enhance gut barrier function by strengthening the integrity of the gut epithelium [23]. Additionally, they restore gut microbiota by competing with pathogenic bacteria and producing antimicrobial compounds [24]. Therefore, various LAB have been suggested as preventive or therapeutic agents for IBD treatment [25,26]. In the present study, we found that oral administration of *L. sakei* CVL-001 alleviated DSS-induced colitis in mice. It not only improved body weight loss and reduced symptoms, such as diarrhea and bloody stools, but also helped recover colon length and histopathological disease severity in the colon. *L. sakei* CVL-001 also regulated the gene expression of pro- and anti-inflammatory cytokines and enhanced the expression of genes related to intestinal barrier functions. Furthermore, we found that *L. sakei* CVL-001 exerts a protective effect against DSS-induced colitis by regulating gut microbiota composition.

Anti-inflammatory cytokines, such as IL-10 and transforming growth factor (TGF)-β, regulate immune response and reduce excessive inflammation. In cases of IBD, they play an important role in controlling excessive immune responses that lead to chronic inflammation [27]. In our study, *L. sakei* CVL-001 administration led to decreased gene expression levels of pro-inflammatory cytokines, such as *IL-1β*, *TNF-α*, and *IL-6*, and increased gene expression levels of anti-inflammatory cytokines, such as *IL-10*, against DSS-induced colitis (Figure 3). These data are consistent with the findings of a previous study, wherein intestinal inflammatory response enhanced the expression of *IL-10* and reduced that of *IL-1β*, *TNF-α*, and *IL-6* against trinitrobenzene sulfonic acid (TNBS)-induced colitis in *L. sakei* S1 administration groups [28]. LAB have several properties that modulate the host’s immune system. Their cell walls are composed of peptidoglycan, lipoteichoic acids, and polysaccharides, which play important roles in the interaction with host immune cells [29]. Moreover, LAB produce a range of anti-inflammatory molecules, including short-chain fatty acids, amines, and bacteriocin [30,31,32]. Previous research has revealed that heat-killed *L. sakei* K040706, which preserve their cell wall components, protect against DSS-induced colitis by suppressing inflammatory mediators such as nitric oxide, TNF-α, IL-1β, and IL-6 [33]. The bioactive product, namely SEL001 from *L. sakei* probio65, which contains various bioactive components such as organic acids, sugar alcohol, and amino acids, exhibits anti-inflammatory effects by regulating TNF-α and IL-6 levels against TNBS-induced colitis [34]. This suggests that several live *L. sakei* CVL-001 components might have contributed to the anti-inflammatory effect against DSS-induced colitis. However, further studies are necessary to identify the key *L. sakei* CVL-001 component that has anti-inflammatory properties.

Intestinal epithelial cells are held together by adherens and tight junctions to maintain their integrity, and secrete mucins to prevent foreign substances, such as bacteria and toxins, from entering the bloodstream [35]. IBD is characterized by destruction of the intestinal epithelium, leading to increased intestinal permeability and translocation of luminal antigens into the bloodstream [36]. Therefore, maintenance of intestinal epithelial integrity through restoring intercellular junctions and mucins in intestinal epithelial cells is considered an important therapeutic goal in IBD treatment [36]. A previous study showed that treatment with *L. sakei* K040706 restored the altered expression levels of genes coding for tight junction proteins, such as ZO-1 and occludin, in DSS-exposed colonic tissue [33]. In our study, histologic evaluation revealed that *L. sakei* CVL-001 administration reversed mucosal damage and integrity disruption caused by DSS treatment (Figure 2). We also found that administration of *L. sakei* CVL-001 led to upregulated expression of genes coding for mucin as well as adherens and tight junction proteins (Figure 4). These results suggest that *L. sakei* CVL-001 administration improves epithelial integrity by regulating the expression of genes coding for adherens and tight junction proteins and mucin.

The co-housing technique is one of the microbiota transfer techniques that utilizes the habits of mice. Mice in the same cage consistently exchange microbiota through coprophagy, grooming, and other social interactions. After co-housing for the period of the experiment, the microbiota composition of two distinct groups will be similar. This technique has been widely accepted to investigate the role of the microbiota in various physiological and pathological conditions [37,38,39]. In our study, we investigated the microbiota-mediated protective effect of *L. sakei* CVL-001 using the co-housing technique. In a separate condition, *L. sakei* CVL-001 administration still improved recovery against DSS-induced colitis. However, in co-housed conditions, the PBS and *L. sakei* CVL-001 treatment groups did not exhibit significant differences in clinical and pathological scoring. These data suggest that the altered microbiota achieved by the co-housing might have contributed to an impaired protective effect compared with the separated group. Taken together, the microbiota modulated by *L. sakei* CVL-001 administration contribute to a protective effect against DSS-induced colitis.

Gut microbiota diversity and composition play an important role in IBD progression. While *L. sakei* WIKIM30 has been demonstrated to restore diversity against 2,4-dinitrochlorobenzene (DNCB)-induced atopic dermatitis, its effects on gut microbiota diversity in animal models of IBD have not been reported [40]. *L. sakei* CVL-001 administration not only increased the alpha diversity of gut microbiota in the *L. sakei* CVL-001-treated group; it also slightly raised that of the gut microbiota in the PBS-treated group, under co-housing conditions (Figure 6A). These data suggest that *L. sakei* CVL-001 administration increases alpha diversity, i.e., the presence of diverse types of bacteria, which is indicative of healthy gut microbiota, to combat DSS-induced colitis.

Gut microbiota comprises two major phyla, namely *Bacteroidetes* and *Firmicutes*, with other relatively minor phyla such as *Proteobacteria* and *Verrucomicrobia*. Dysbiosis, which is an imbalance in the gut microbiota, is strongly related to numerous diseases [6]. The *Firmicutes* to *Bacteroidetes* (F/B) ratio of gut microbiota has been suggested as a biomarker of dysbiosis. For example, the F/B ratio of gut microbiota in obese individuals was observed to be higher than that in healthy individuals [41]. Additionally, alteration of the F/B ratio associated with increased disease activity has been reported in patients with IBD [9,42]. In the previous studies, *L. sakei* has been shown to affect the F/B ratio of gut microbiota. For instance, the gut microbiota of mice treated with *L. sakei* K040706 to combat DSS-induced colitis showed a decreased F/B ratio [33]. Additionally, compared with the control group, the *L. sakei* ADM14-treated group demonstrated a lower F/B ratio against the high-fat diet-induced obesity mouse model [43]. Similarly, our results showed decreased F/B ratio in *L. sakei* CVL-001-treated mice than did the PBS-treated group, regardless of co-housing (Figure 6C). While further research is required to fully understand the effects of *L. sakei* on F/B ratio, the available evidence suggests that they can increase the abundance of *Bacteroidetes* and decrease that of *Firmicutes* to protect against microbiota-related diseases.

*Proteobacteria* include pathogenic genera, such as *Escherichia*, *Salmonella*, and *Proteus*, which affect IBD’s severity [44,45]. Increased abundance of *Proteobacteria* in the gut represents an unbalanced alteration of the gut environment [45]. Previous research has demonstrated that administration of *L. sakei* reduces the abundance of *Proteobacteria* in animal models of IBD. Fecal analysis of DSS-induced mice showed that *L. sakei* K040706 administration suppresses the abundance of *Proteobacteria* [33]. Furthermore, a significant reduction in pathogenic *Escherichia-Shigella* levels has been observed in the *L. sakei* K040706 administration group [33]. Administration of the bioactive product, namely SEL001 from *L. sakei* probio65, decreases the abundance of *Proteobacteria* against TNBS-induced colitis [34]. In this study, *L. sakei* CVL-001 administration led to decreased abundance of *Proteobacteria* at the phylum level and decreased abundance of *Escherichia* and *Proteus* spp. at the genus level (Figure 6C). *Escherichia* and *Proteus* spp., which are typically considered pathogens belonging to the *Proteobacteria* phylum, have been found to serve as biomarkers for colitis severity, and to promote colonic inflammation and mucosal bacterial dysbiosis in mice [46,47]. Therefore, administration of *L. sakei* CVL-001 inhibits the colonization of pathogenic bacteria, thereby providing a protective effect against DSS-induced colitis. In our study, we performed our microbiota analysis based on OTUs that have been widely used in previous studies for comparative analysis with other DSS-induced colitis experiments. However, with the development of algorithmic techniques, ASV based on analysis may provide a significant advantage for the more precise identification of microbes. To unravel the relationship between gut microbiota and IBD, there is a need to accumulate microbiota data using ASV analysis for further studies.

In conclusion, administration of *L. sakei* CVL-001 could effectively ameliorate DSS-induced colitis by inhibiting excessive inflammatory response, maintaining epithelial integrity, and modulating gut microbiota. Therefore, *L. sakei* CVL-001 may be a potential candidate for the treatment of IBD.

## Figures and Tables

**Figure 1 microorganisms-11-01359-f001:**
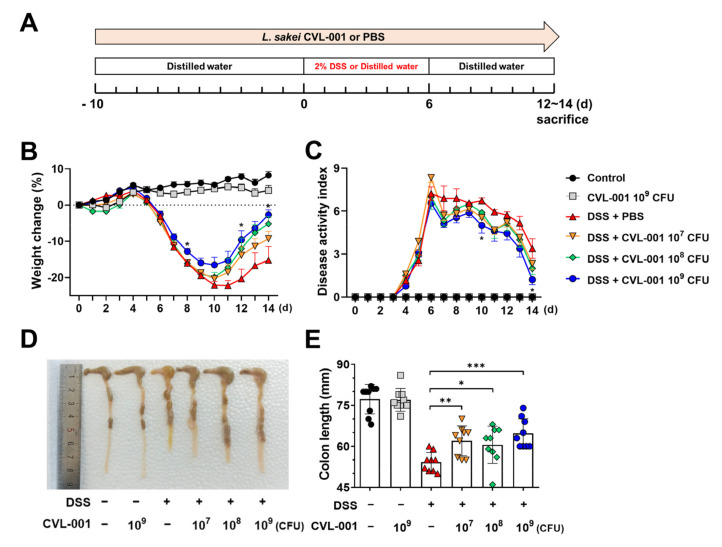
Protective effects of *L. sakei* CVL-001 against DSS-induced colitis. (**A**) Scheme of the experimental design. Mice were randomly divided into six groups (*n* = 9/group). (**B**) Body weight changes (%) during the colitis and recovery phases (mean ± SEM) and total weight loss relative to day 0. Statistical indicators (*) were compared with the DSS + PBS group. (**C**) Disease activity index of mice (mean ± SEM). Statistical indicators (*) were compared with the DSS + PBS group. (**D**) Macroscopic appearance of representative colons from each group. (**E**) Quantification of colon length (mean ± SEM). The significance of differences among the DSS-treated groups was assessed using a two-way analysis of variance (ANOVA) for body weight and disease activity index and a Mann–Whitney U test for colon length, with the level of significance set at *p* < 0.05 (*), *p* < 0.01 (**), and *p* < 0.001 (***). SEM, standard error of mean; DSS, dextran sulfate sodium; PBS, phosphate-buffered saline.

**Figure 2 microorganisms-11-01359-f002:**
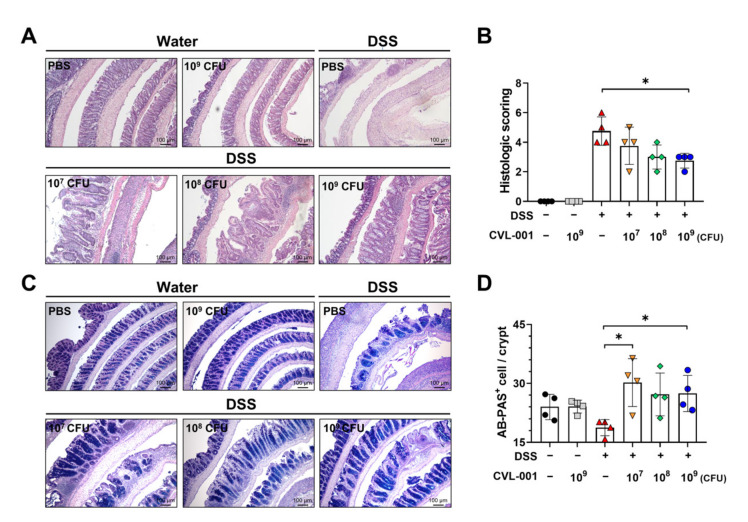
Protective effects of *L. sakei* CVL-001 on intestinal histological features. (**A**) Microscopic appearance of representative colons from each group (H&E staining). (**B**) Histologic scoring of the colon (mean ± SEM). (**C**) Microscopic appearance of representative colons from each group (AB-PAS staining). (**D**) Number of AB-PAS-positive cells/colonic crypt. The significance of differences among the DSS-treated groups was assessed using the Mann–Whitney U test, with the level of significance set at *p* < 0.05 (*). H&E, hematoxylin and eosin; AB-PAS, Alcian blue–periodic acid Schiff; SEM, standard error of mean; DSS, dextran sulfate sodium.

**Figure 3 microorganisms-11-01359-f003:**
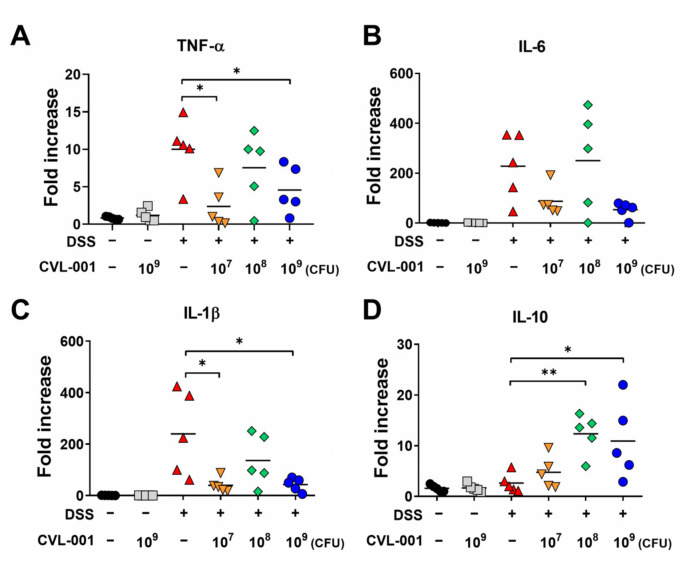
Evaluation of mRNA expression of cytokines in the colons of mice with DSS-induced colitis. mRNA extracted from colonic tissue was prepared for real-time PCR. The mRNA levels of (**A**) *TNF-α*, (**B**) *IL-6*, (**C**) *IL-1β*, and (**D**) *IL-10* were evaluated using each primer and normalized against GAPDH (*n* = 5). The significance of differences among the DSS-treated groups was assessed using the Mann–Whitney U test, with the level of significance set at *p* < 0.05 (*) and *p* < 0.01 (**). PCR, polymerase chain reaction; TNF, tumor necrosis factor; IL, interleukin; GAPDH, glyceraldehyde 3-phosphate dehydrogenase; SEM, standard error of mean; DSS, dextran sulfate sodium.

**Figure 4 microorganisms-11-01359-f004:**
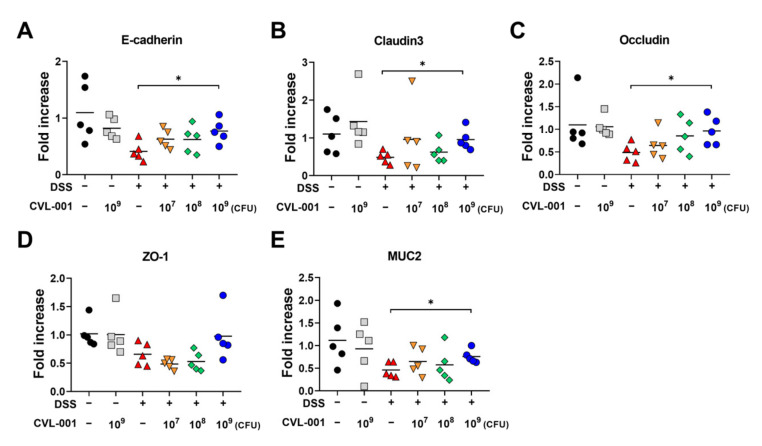
Evaluation of mRNA expression of genes coding for adherens junction protein, tight junction protein, and mucin in the colons of mice with DSS-induced colitis. mRNA extracted from colonic tissue was prepared for real-time PCR. mRNA levels of genes coding for (**A**) E-cadherin, (**B**) claudin3, (**C**) occludin, (**D**) ZO-1, and (**E**) MUC2 proteins were evaluated using each primer and normalized against GAPDH (*n* = 5). The significance of differences among the DSS-treated groups were assessed using the Mann–Whitney U test, with the level of significance set at *p* < 0.05 (*). PCR, polymerase chain reaction; ZO, zonula occludens; MUC, mucin; GAPDH, glyceraldehyde 3-phosphate dehydrogenase; DSS, dextran sulfate sodium.

**Figure 5 microorganisms-11-01359-f005:**
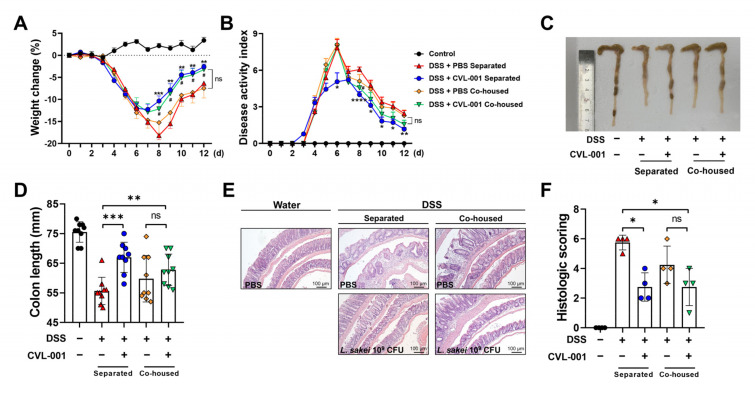
Microbiota modulation effects of *L. sakei* CVL-001 administration under co-housing conditions. Mice were randomly divided into five groups (*n* = 9/group). (**A**) Body weight changes (%) in mice during the colitis and recovery phases (mean ± SEM) and total weight loss relative to day 0. Statistical indicators (*, #) were compared with the DSS + PBS separated group. (**B**) Disease activity index of mice (mean ± SEM). Statistical indicators (*, #) were compared with the DSS + PBS separated group. (**C**) Macroscopic appearance of representative colons from each group. (**D**) Quantification of colon length (mean ± SEM). (**E**) Microscopic appearance of representative colons from each group. (**F**) Histologic scoring of colon (mean ± SEM). The significance of differences among the DSS-treated groups was assessed using a two-way analysis of variance (ANOVA) for body weight and disease activity index and a Mann–Whitney U test, with the level of significance set at *p* < 0.05 (*, #), *p* < 0.01 (**), *p* < 0.001 (***), and *p* < 0.0001 (****). SEM, standard error of mean; DSS, dextran sulfate sodium; PBS, phosphate-buffered saline.

**Figure 6 microorganisms-11-01359-f006:**
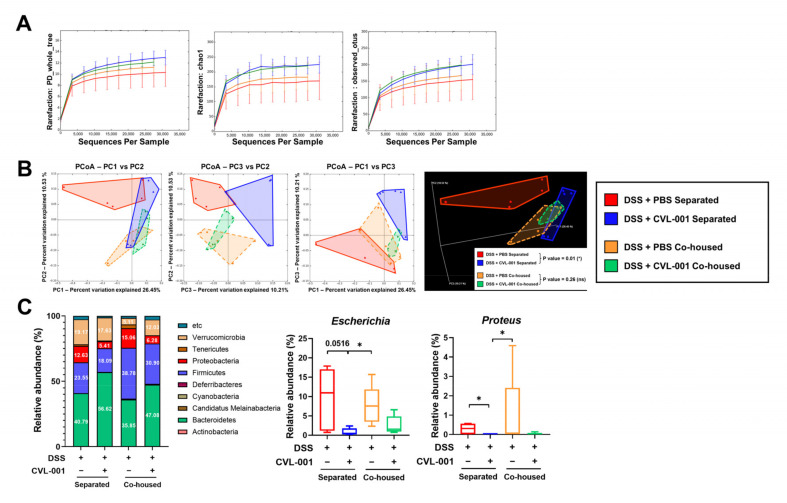
Microbiota analysis of feces of mice with DSS-induced colitis. (**A**) Rarefaction curve of alpha diversity of the gut microbiota showing index for the PD_whole_tree and Chao1, observed OTUs (*n* = 5). (**B**) Principal component analysis (PCoA) illustrating beta diversity of each group using 2D and 3D plots (unweighted UniFrac distance, *n* = 5). (**C**) Abundance of microbiota composition at the phylum and genus level. The *p* values were determined using PERMANOVA on beta diversity, with the level of significance set at *p* < 0.05 (*). The significance of differences among the DSS-treated groups was assessed using the Mann–Whitney U test, with the level of significance set at *p* < 0.05 (*) for the microbiota composition analysis. OTU, operational taxonomic unit; PD, phylogenetic diversity; DSS, dextran sulfate sodium.

**Table 1 microorganisms-11-01359-t001:** Scoring system for the disease activity index.

Score	Weight Loss	Stool Consistency	Blood
**0**	None	Normal	Normal
**1**	1–5%	Soft but still formed	Slight hemoccult
**2**	6–10%	Soft	Positive hemoccult
**3**	11–18%	Wet, very soft	Blood traces in stool
**4**	>18%	Watery diarrhea	Gross rectal bleeding

**Table 2 microorganisms-11-01359-t002:** Scoring system for histological evaluation.

Score	Epithelial Integrity	Lamina Propria Inflammatory Cell Infiltration
**0**	None	Infrequent
**1**	Focal epithelial damage	Increased, some neutrophils
**2**	Mucosal erosions and ulceration	Submucosal presence of inflammatory cell clusters
**3**	Extensive epithelial damage	Transmural cell infiltration

**Table 3 microorganisms-11-01359-t003:** Primer sequences used in this study.

Genes	Sequence 5′-3′
**IL-6**	**F**: GCT ACC AAA CTG GAT ATA ATC AGG A
	**R**: CCA GGT AGC TAT GGT ACT CCA GAA
**TNF-α**	**F**: TCT GTC TAC TGA ACT TCG GGG TGA
	**R**: TTG TCT TTG AGA TCC ATG CCG TT
**IL-1β**	**F**: CAA CCA ACA AGT GAT ATT CTC CAT G
	**R**: ATC CAC ACT CTC CAG CTG CA
**IL-10**	**F**: TAA CTG CAC CCA CTT CCC AG
	**R**: AGG CTT GGC AAC CCA AGT AA
**GAPDH**	**F**: CGG AGT CAA CGG ATT TGG TCG TAT
	**R**: AGC CTT CTC CAT GGT GGT GAA GAC
**E-cadherin**	**F**: CAG GTC TCC TCA TGG CTT TGC
	**R**: CTT CCG AAA AGA AGG CTG TCC
**Occludin**	**F**: CAC ACT TGC TTG GGA CAG AG
	**R**: TAG CCA TAG CCT CCA TAG CC
**Claudin-3**	**F**: AAC TGC GTA CAA GAC GAG ACG
	**R**: ATC CCT GAT GAT GGT GTT GG
**ZO-1**	**F**: CCT AAG ACC TGT AAC CAT CT
	**R**: CTG ATA GAT ATC TGG CTC CT
**MUC2**	**F**: TGC TGA CGA GTG GTT GGT GAA TG
	**R**: TGA TGA GGT GGC AGA CAG GAG AC

## Data Availability

The data that support the finding of this study are available from the corresponding author upon reasonable request.

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
