# Peer review of "Oral Administration of Lactobacillus sakei CVL-001 Improves Recovery from Dextran Sulfate Sodium-Induced Colitis in Mice by Microbiota Modulation"

_microorganisms, 2023, doi:10.3390/microorganisms11051359_

Round 1
Reviewer 1 Report
The subject of the manuscript is interesting, but the authors should clarify some aspects to improve the work.
In line 63 the word kimchi is in italics, homogenize throughout the manuscript, removing the italics.
It is not clear why the authors used in line 90 three concentrations of the probiotic, and in line 97 they use only one.
The description of the groups is not clear, since in line 89 they talk about 5 and 6 groups, which ones were they?
The use of separated or co-housing is not clear.
It is not clear the results by the co-housed experiment, gene expression by mRNA is not observed.
What dose of microorganism was used for the microbiota analysis, both for the separated and co-housed experiment?
The authors do not discuss any of the results obtained from the co-housed experiment. It is not clear for what purpose they performed this experiment
Reviewer 2 Report
This study shows that Lactobacillus sakei CVL-001 can protect mice from DSS-induced colitis by regulating the immune response and intestinal integrity via gut microbiota modulation. The findings suggest that Lactobacillus sakei CVL-001 may have therapeutic potential for IBD treatment. Overall, this study provides valuable insights into the relationship between gut microbiota and IBD. I only have some minor comments.
Line 81. Only females were used in this study, any particular reasons? As female is resistant to DSS treatment, the male is more widely used for the colitis mouse model. Please clarify.
Line 144. QIIME, version 1.9 is too old, it would be beneficial to use the latest version that uses the ASV instead of OUT.
Line 219. Do you have any protein data?
Line 271-273. What is the P-value? Any statistical differences for the beta diversity?
Round 2
Reviewer 1 Report
the authors responded satisfactorily to the comments